# Bronchial Physiological Saline Injection to Visualize Peripheral Pulmonary Lesions in Ultrathin Bronchoscopy

**DOI:** 10.3390/diagnostics15233029

**Published:** 2025-11-28

**Authors:** Mika Nakao, Tamio Okimoto, Noriaki Kurimoto, Ryosuke Tanino, Misato Kobayashi, Kazuhisa Nakashima, Takamasa Hotta, Yukari Tsubata, Takeshi Isobe

**Affiliations:** Division of Medical Oncology & Respiratory Medicine, Department of Internal Medicine, Shimane University Faculty of Medicine, 89-1 Enya-cho, Izumo-shi, Shimane 693-8501, Japan

**Keywords:** ultrathin bronchoscopy, peripheral pulmonary lesions, saline immersion, lung cancer diagnosis, narrow-band imaging, white-light imaging

## Abstract

**Background:** During bronchial lavage, the lumen can be viewed better by injecting physiological saline into the bronchus. This study aimed to investigate the effect of injecting physiological saline into the bronchus while directly observing the bronchial lumen with peripheral pulmonary lesions using an ultrathin bronchoscope. **Methods:** We evaluated changes in the depth of field and signal-to-noise ratio due to saline immersion in bronchoscopic images captured with line patterns. We injected physiological saline through the working channel into the bronchial lumen and directly observed the lesions (mainly in cases of suspected peripheral lung cancer) in 38 patients using white-light and narrow-band imaging. Ultrathin bronchoscopic findings and histopathological diagnoses were analyzed. **Results:** Saline immersion extended the range of subject distance within the depth of the field with an improved signal-to-noise ratio. Saline immersion elevated the signal-to-noise ratio and increased the highest signal-to-noise ratio in the white-light images from 3.12 dB to 3.77 dB (1.21-fold). Under narrow-band imaging, saline immersion also increased the highest signal-to-noise ratio from 5.18 dB to 6.44 dB (1.24-fold). We were able to directly observe the bronchial lumen of the peripheral lesions in 30/38 cases (78.9%). Three (60%) of the five squamous cell carcinomas were of the epithelial type, and 12 (92%) of the 13 adenocarcinomas, including peripheral pulmonary lesions, were of the subepithelial type, similar to central lesions. **Conclusions:** Injecting saline into the peripheral bronchus is useful for direct bronchoscopic evaluation of peripheral lesions.

## 1. Introduction

Flexible bronchofiberscopes were developed by Ikeda [1] in the 1960s. Bronchoscopic approaches for the diagnosis of peripheral pulmonary lesions have evolved substantially over the past six decades. Conventional fluoroscopy has been supplemented by more advanced modalities, including radial probe endobronchial ultrasonography [2], which delineates the spatial relationship between peripheral lesions and adjacent bronchi, as well as guide sheath techniques [3] and virtual bronchoscopy [4,5,6] to facilitate accurate bronchial navigation. The recent expansion of robotic bronchoscopy [7], particularly in the United States and Europe, further underscores the shift toward more precise transbronchial access.

Concurrently, reductions in bronchoscope diameter have enabled deeper airway insertion. Although thin bronchoscopes (outer diameter ~4 mm) improved peripheral accessibility, direct visualization of intraluminal features within bronchi leading to peripheral lesions remained challenging. With the increasing use of ultrathin bronchoscopes (outer diameter ~3 mm), it has become possible in select cases to directly observe bronchial luminal structures at the lesion site.

Moreover, during bronchoalveolar lavage for diffuse lung disease, we observed that saline instillation improved visualization of longitudinal mucosal folds by stabilizing and distending the airway lumen. These observations led us to hypothesize that saline instillation combined with ultrathin bronchoscopy may enable direct visualization of bronchial luminal characteristics within peripheral pulmonary lesions.

Bronchoscopic findings are classified in the *General Rule for Clinical and Pathological Record of Lung Cancer* published by the Japan Lung Cancer Society [8] Tanaka et al. [9] evaluated the luminal findings of the peripheral bronchus using an ultrathin bronchofiberscope; however, they did not compare direct bronchofiberscopic findings with histological type. To the best of our knowledge, the relationship between the histological type and luminal findings of the peripheral bronchus using an ultrathin bronchoscope has not been discussed.

In addition, we hypothesized that saline injections would be useful for bronchial lumen observation. When observing bronchial lumen using an ultrathin bronchoscope, injecting physiological saline (saline) into the peripheral bronchus through the working channel can widen the bronchial lumen, making it easier to focus on the target. A future diagnostic pathway is expected to be established in which the intraluminal findings of bronchi containing peripheral pulmonary lesions are directly observed using an ultrathin bronchoscope, enabling real-time differentiation between benign and malignant lesions and facilitating targeted biopsies under direct vision. Visualization of the bronchial lumen under saline immersion may enhance the clarity of mucosal and vascular structures, thereby improving diagnostic accuracy for peripheral pulmonary lesions.

Therefore, in this study, we aimed to conduct an experiment to evaluate the resolution of an ultrathin bronchoscope under air and saline immersion using a chart by injecting saline through the working channel into the bronchial lumen.

## 2. Materials and Methods

### 2.1. Experimental Procedures

A bronchoscopic system comprising an ultrathin bronchoscope (BF-XP260F; Olympus Medical Systems, Tokyo, Japan), a video processor (CV-260SL; Olympus Medical Systems), a light source (CLV-260SL; Olympus Medical Systems), and a video recorder (IMH-10; Olympus Medical Systems) was used to record bronchoscopic videos capturing a line pattern comprising 4.49 line pairs/mm (Group 2, Element 2) on a 2” × 2” Positive, 1951 USAF Resolution Target (Edmund Optics, Barrington, NJ, USA). Line pattern images for different subject distances were extracted from recorded videos. The distance between the end of the insertion cord and the target was determined using a Manual XYZ Stage B09–27 (Suruga Seiki, Shizuoka, Japan) by measuring the length between the vertical stage and the stage base using a Vernier caliper 530-100 (Mitutoyo, Kawasaki, Japan). ImageJ 1.53c (National Institutes of Health, Bethesda, MD, USA) was used for image quantification. All images were cropped manually (only the target’s line pattern) and were standardized by resizing to the same width as that of the image captured at a distance of 0.5 mm in saline using ImageJ Resize (depth = 1, constraint average interpolation = bicubic) and cropping one-fourth of the upper and lower sides of the image as the evaluation area. Each image was set in an 8-bit grayscale format for subsequent use.

The binary image was generated using the ImageJ Threshold with the 0–50th percentile of brightness as the evaluation image. The dark pixels in the image were masked by adding a binary image to the evaluation image. Lines in the mask were separated and considered within the depth of the field for a 111-μm resolution.

The signal-to-noise ratio (SNR) of the obtained images was measured as follows. The gray value of the evaluation area image was measured using the ImageJ Plot Profile with High-Resolution Plot (fixed minimum = 0, maximum = 255). Four vertical lines with equal intervals were plotted manually on the gray-value graph to sandwich two concave white signals and a convex black signal. The areas of the signal (S1, S2, and S3) and noise (N1, N2, and N3) were measured. The SNR was calculated using the following formula:SNR = 10 × log_10_ [{(S1 + S2)/2 + S3}/{(N1 + N2)/2 + N3}] (dB).

### 2.2. Peripheral Bronchial Lumen Observation

Forty patients were examined prospectively using an ultrathin bronchoscope before undergoing endobronchial ultrasonography using a guide sheath (EBUS-GS) with a thin bronchoscope (BF-P260F; outer diameter, 4.0 mm; channel diameter, 2.0 mm; Olympus Medical Systems) from May to September 2018 at our institution. The eligibility criteria were patients with peripheral pulmonary lesions who were scheduled to undergo bronchoscopy (irrespective of the suspected disease type before bronchoscopy) and patients in whom the bronchi (inner diameter < 4 mm) through which the lesion was reached were identified using high-resolution computed tomography (HR-CT). This study was approved by the appropriate Ethics Committee of the Shimane University Faculty of Medicine (approval number: 3154) and registered with the UMIN Clinical Trial Registry (UMIN000033626). Written informed consent was obtained from all patients.

Bronchoscopists identified the bronchi to reach the lesion and used HR-CT to prepare a bronchial diagram. The lobar bronchus was the first generation according to the *General Rule for Clinical and Pathological Record of Lung Cancer* published by the Japan Lung Cancer Society [8]. An ultrathin bronchoscope was guided toward the lesion based on the bronchial diagram. Once the bronchoscope reached the lesion, distant and close-up images were captured using white light and narrow-band imaging (NBI) conditions. Then, 1 mL of saline was intermittently injected through the working channel; the injection was discontinued when the bronchial lumen was filled with saline, and distant and close-up images were taken again. The ultrathin bronchoscope was advanced as far as possible until resistance was encountered, and saline was injected at that position. The bronchoscope was not advanced further after the saline injection. We classified the bronchial lumen into epithelial and subepithelial types, clear and unclear borders, and flat, raised, and obstructive forms. An “epithelial-type lesion” is characterized by an irregular epithelium lacking a smooth, glossy surface, whereas a “subepithelial-type lesion” presents with a regular, smooth, and glossy surface. We classified the borders of lesions as “clear” when a distinct boundary between the normal bronchus and the lesion could be traced, and as “unclear” when the boundary could not be clearly followed. The form of the lesion was categorized as “flat” when the bronchial inner diameter showed no obvious difference compared with that of the normal bronchus, as “raised” when the lesion projected inward and the bronchial inner diameter at the lesion site was smaller than that of the normal bronchus, and as “obstructive” when the lesion completely obstructed the bronchial lumen. After inspection of the bronchial lumen, the lesions were brushed using a cytology brush (BC-203D-2006, outer diameter 1.1 mm; Olympus Medical Systems).

After ultrathin bronchoscopic examination, we advanced a thin bronchoscope to the lesions and inserted a radial-type ultrasonic probe (UM-S20-17S; Olympus Medical Systems) with a guide sheath (SG-200C; Olympus Medical Systems) through the working channel. The location of the lesions was confirmed using both endobronchial ultrasonography and fluoroscopy. We performed brushing and forceps biopsy several times using a guide sheath.

Cytological and histopathological diagnoses were performed by a pathologist. In our study, “suspicious” findings were considered non-malignant based on cytological or histopathological findings. For cases in which cytological and histopathological diagnoses were not confirmed by bronchoscopy, the final diagnosis was established by microbiological examination and pathological diagnosis by repeat bronchoscopy or surgery.

The primary endpoint was the number of patients in whom the bronchial lumen could be directly visualized with an ultrathin bronchoscope. Secondary endpoints included bronchoscopic findings, assessment of histopathological type, the bronchial generation reached by ultrathin bronchoscopy, and bronchoscopic findings with and without saline injection.

Two pulmonologists judged whether it was possible to evaluate ultrathin bronchoscopic findings (unevenness of the bronchial luminal surface, circular and/or longitudinal folds, blood vessels of the subepithelium, and perspective) using white light and NBI under air or saline immersion on the recorded video without patient information.

### 2.3. Statistical Analysis

All statistical analyses were performed using R version 4.0.3. [10] The exact 2 × 2 package for R [11] was used to perform the exact McNemar’s test. Results were considered statistically significant at *p* < 0.05.

### 2.4. Sample Size

Because no previous studies have reported the relationship between ultrathin bronchoscopic intraluminal findings and histopathological types, a formal sample size calculation was not feasible. Therefore, this study was designed as an exploratory investigation. Based on the number of cases of solitary peripheral lung nodules diagnosed at Shimane University Hospital in 2017, we estimated that approximately 40 cases could be prospectively enrolled within one year and set this number as the target sample size.

## 3. Results

### 3.1. Bronchoscopic Evaluation

The bronchial lumen was easier to observe in saline immersion than in air (Figure 1A). Thus, we obtained images including a line pattern with 4.49 line pairs/mm at different distances to the target for 111-μm resolution, considering the tumor vessel diameter (Figure 1B,C) [12]. The bronchoscopic image was cropped and evaluated, avoiding reflected light and shadows of black lines due to light emission beside the lens (Figure 1D,E).

### 3.2. Compartmentalization of Line Pixels

The size of the target in the bronchoscopic image increased by approximately 1.33 times under saline immersion (Figure 2A). The masked lines of the images obtained with and without saline were observed using white light (Figure 2B,C) and NBI (Appendix A). Saline immersion increased the range of subject distance within the depth of field in the direction away from the target under both white-light and NBI conditions (Figure 2D).

### 3.3. SNR Evaluation of Bronchoscopic Images

We measured the SNRs of the obtained images (Appendix A). Overall, saline immersion elevated the SNRs and increased the highest SNR in the images obtained using white-light from 3.12 dB to 3.77 dB (1.21-fold). However, the distance with the highest SNR shifted from 3.0 mm to 4.5 mm (Figure 3). Using NBI, saline immersion also increased the highest SNR from 5.18 dB to 6.44 dB (1.24-fold), and the distance with the highest SNR shifted from 3.0 mm to 4.5 mm, the same as when white light was used (Figure 3).

### 3.4. Peripheral Bronchial Findings

Forty patients were enrolled in this study (Figure 4). Two patients were not examined according to the established protocol because a thin bronchoscope could not be used; therefore, 38 patients were examined according to the protocol. In eight patients, we could not directly observe the peripheral lesion because the lesion was not reachable, or the bronchial lumen was occupied by sputum or a clot. Using an ultrathin bronchoscope, bronchoscopic observation of the peripheral lesion was successful in 78.9% of patients (30/38).

We excluded four patients. In three patients, a definite diagnosis could not be obtained. One patient was excluded because the bronchoscopy was performed during ongoing chemotherapy, which might have affected the bronchoscopic findings [13].

And chemotherapy could have altered the bronchoscopic findings in one patient), a definite diagnosis was confirmed in 26 of the 38 patients (68.4%), and the bronchoscopic findings were compared with the histopathological diagnosis (Table 1). Of the 26 patients, the final diagnosis was confirmed by bronchoscopy in 20 patients, bronchoscopy was repeated in two patients, and surgery after bronchoscopy without a confirmed diagnosis was performed in four patients. The median number of bronchial generations close to the peripheral lesion on HR-CT images was six. The median number of generations reached by the ultrathin bronchoscope was also 5.5, whereas that reached by the thin bronchoscope was four.

The border between the normal bronchial area and the lesion was unclear in the subepithelial type (Table 2). Of the five squamous cell carcinomas (SCCs), three were epithelial, two were subepithelial, two were flat, two were raised, and one was obstructive. Of the 13 adenocarcinoma lesions, one was epithelial and 12 were subepithelial; eight were flat, four were raised, and one was obstructive. Even for peripheral pulmonary lesions, adenocarcinomas were of the subepithelial type (92.3%, 12/13 cases), whereas SCCs were of various types. Representative cases of epithelial and subepithelial types are shown in Figure 5 and Figure 6.

As shown in Figure 5, the left B^9^aiiα was reached (Figure 5A–C), confirming the histopathological diagnosis of SCC (Figure 5D). In air, the ultrathin bronchoscope showed an unclear image of B^9^aiiα under white light (Figure 5E). However, it showed a whitish, irregular surface in saline immersion when white light was used (Figure 5F) and an irregular epithelium when NBI was used (Figure 5G).

As shown in Figure 6, the right B^6^biiβxx was obtained (Figure 6A,B), confirming the histopathological diagnosis of adenocarcinoma (Figure 6C). Using white light, the ultrathin bronchoscope showed an unclear image of B^6^biiβxx in air (Figure 6D) and a whitish subepithelial lesion after saline injection (Figure 6E), whereas when NBI was used, it was a whitish subepithelial lesion with an abrupt caliber change in the vessel (Figure 6F).

Two pulmonologists evaluated the ultrathin bronchoscopic findings obtained using white light (26 cases with definite diagnosis) and NBI (11 cases with definite diagnosis; 15 cases recorded as data-deficient) in air and saline immersion on the recorded video. In all findings by both raters, the number of distinguishable cases increased after saline injection into the bronchus under white light (Table 3). In contrast, in NBI, Rater 1 was able to observe the vessels significantly in saline immersion, whereas Rater 2 tended to observe the vessels more easily in saline immersion. However, no significant differences were observed (Appendix A).

### 3.5. Complications

No adverse events were observed with bronchial saline injections.

## 4. Discussion

In this study, we demonstrated the usefulness of saline injection in improving the SNR of bronchoscopic images, confirming that the peripheral bronchus can be directly observed with the aid of saline injection.

The bronchial lumen is easier to observe when injected with saline than when filled with air. In our experience, saline clears mucus, provides a clean bronchoscopic view, suppresses lens flares, deepens the depth of field, and makes it easier to observe the bronchial lumen. The bronchus is usually observed in air; however, it is difficult to sufficiently observe the peripheral bronchial lumen in air. The usefulness of saline injections is well known in the field of gastrointestinal endoscopy. Yao et al. [14] reported the advantages of the water immersion technique as follows: (1) the removal of light reflection and lens flare by water makes it easy to observe micro-vessel construction and mucosal surface microstructure; (2) the depth of observation becomes deeper at maximum magnification; and (3) the resolution tends to increase. Additionally, the effects of saline on bronchoscopy have recently been reported. Morikawa et al. [15] reported that endobronchial ultrasonography can be performed by injecting saline into the peripheral cavitary lesions. To the best of our knowledge, quantitative evaluation of the effects of saline injections on bronchoscopy is lacking.

This study demonstrated that saline injection during bronchoscopy extended the range of the subject’s distance within the depth of the field and improved image quality. In liquid immersion microscopy, a high-refractive medium filling the object space increases the apparent numerical aperture [16]. This effect is demonstrated by saline with a higher refractive index (n_S_ = 1.33) than air (n_A_ = 1.00), narrowing the angle of view by approximately three-fourths at the same subject distance. Saline increased the magnification (1.33-fold) and eventually improved the spatial resolution, resulting in an increased SNR under both white light (1.21-fold) and NBI (1.24-fold). Moreover, saline increased the distance to the target with the highest SNR from 3.0 mm to 4.5 mm (1.5-fold), indicating a focal length increase. The focal lengths of the thin lenses without saline (f_A_) and with saline (f_S_) were calculated as [17]
$$-\frac{n_{D}}{f_{A}}=\frac{n_{L}-n_{A}}{r_{1}}+\frac{n_{D}-n_{L}}{r_{2}}$$
$$-\frac{n_{D}}{f_{S}}=\frac{n_{L}-n_{S}}{r_{1}}+\frac{n_{D}-n_{L}}{r_{2}}$$ where n_L_ is the refractive index of the lens, n_D_ is the refractive index difference between the lens and photodetector, r_1_ is the radius of curvature of the lens surface facing the subject, and r_2_ is the radius of curvature of the lens surface facing the photodetector. Although we could not obtain detailed specifications for the optical components of the bronchoscope, if the lens is biconvex with the same radius of curvature (r_1_ = −r_2_), n_L_ = 1.5 as a glass lens, and n_D_ = n_A_ = 1.00 as air, then the focal length increase ratio is f_S_/f_A_ = 1.49. This is consistent with the increased ratio (1.5-fold) observed in this study. Despite the increase in focal length, we did not observe any drawbacks for saline injection in the range of distances to the target (0.5–10 mm) under global SNR improvement.

We observed the bronchus and subsequent bronchial generations using an ultrathin bronchoscope. The distance from the bronchus to the next bronchial generation in the proximal region of the terminal bronchiole was reportedly approximately 0.5–1 cm [18]. Our experimental findings indicate the usefulness of bronchial saline injection due to the SNR improvement in the range of distance (0.5–10 mm). In addition, we confirmed whether saline injection had any adverse effects by examining the raters’ evaluations. The evaluation of observations under white light revealed that the observers were able to obtain more findings using saline injection than air (Table 3). Thus, saline does not hinder observation under white light or NBI. Furthermore, in addition to its optical effects, saline also acts physically by displacing mucus, thereby enabling direct visualization of the bronchial lumen. This effect may be particularly beneficial when performing bronchoscopy in patients with chronic bronchitis. Moreover, saline injection takes only a few seconds, and we believe it does not prolong the total examination time. Therefore, we propose the observation of peripheral pulmonary lesions under bronchial saline injection using an ultrathin bronchoscope.

Regarding the endoscopic findings of the peripheral airways using an ultrathin bronchoscope, Tanaka et al. [3] reported abnormal findings of the bronchial wall (reddening, pallor, absence of mucosal surface luster, swelling, engorgement of blood vessels, irregular mucosal surface, and polypoid lesion), endobronchial abnormalities (stenosis, obstruction, and ectasis), and abnormal substances in the bronchial lumen (secretion and pigmentation). Watanabe [19] and Ohsaki [20] also reported that the mucosal invasive type is usually observed in hilar-type SCC, and the submucosal invasive type is mainly observed in peripheral adenocarcinoma, SCC, large cell carcinoma, and small cell carcinoma. However, the relationship between peripheral bronchial findings and histological type has not yet been investigated. In our study, we suspected that the border between lesions and the normal bronchus would be clear for epithelial-type lesions, whereas the border would be unclear for subepithelial-type lesions. Even peripheral pulmonary lesions, similar to central lesions, adenocarcinomas are of the subepithelial type, whereas SCCs are of various types. When an ultrathin bronchoscope reached the inside of the SCC lesion, we directly observed the exposed part of the epithelium. In contrast, when the ultrathin bronchoscope reached only the margin of the SCC lesion, we occasionally observed a subepithelial type extending under the epithelium. Therefore, we suspected that various luminal findings were present in the SCC.

Recently, although there have been some case reports on the usefulness of NBI in bronchoscopy [12,15,21,22,23], there are only a few summaries of NBI findings when observing peripheral bronchial lesions. Shibuya et al. [22] and Zaric et al. [23] reported that dotted, tortuous, and abruptly ending vessels suggest malignancy, especially SCC. Even in peripheral lesions, observing the vasculature using NBI and white light may be useful for simulating histological types. In our laboratory experiments, when using NBI, saline injection improved the image quality of the target for 111-μm^6^ resolution, considering the tumor vessel diameter, suggesting that saline injection is also useful in NBI observations (Figure 3). The improvement in the observation ability of the saline injection varied depending on the rater. NBI with saline injection tended to make the blood vessels easier to observe. In NBI observations, saline injection was considered to have almost no adverse effects (Appendix A). The diagnostic yields of ultrathin bronchoscopy and EBUS-GS were reportedly 60–74% [24,25,26] and 59–77%, respectively [25,27,28]. In our study, observation of the bronchus leading to the peripheral lesion using the ultrathin bronchoscope was successful in 78.9% of patients (30/38) (Figure 4). When malignancy is strongly suspected based on bronchoscopic findings, the diagnostic yield of lung cancer can be further improved if a biopsy can be performed from a site with abnormal findings under direct vision. By including observational findings, such as NBI with saline injection, it may be possible to further improve the diagnostic yield. Even for peripheral lesions, we believe that it is important to closely observe the bronchus in the same manner as for central lesions.

The limitations of this study are as follows. The sample size was small because the study was conducted at a single institution. In addition, pure ground-glass nodules were difficult to detect using endobronchial ultrasonography after saline was injected into the bronchus around the lesion; therefore, only solid lesions were analyzed in our study. This study could not evaluate whether saline injection improved the cytological or histological diagnostic yield. Further comparative studies are therefore required to clarify its diagnostic utility. In addition, newly developed devices such as the Iriscope [29,30,31] and SUKEDACHI™ [32,33] warrant future comparative evaluation to determine their relative advantages.

## 5. Conclusions

In conclusion, injecting saline into the peripheral bronchus is useful for direct bronchoscopic observation in both quantitative experiments and clinical studies. Ultrathin bronchoscopic findings indicate the histological type in some cases.

## Figures and Tables

**Figure 1 diagnostics-15-03029-f001:**
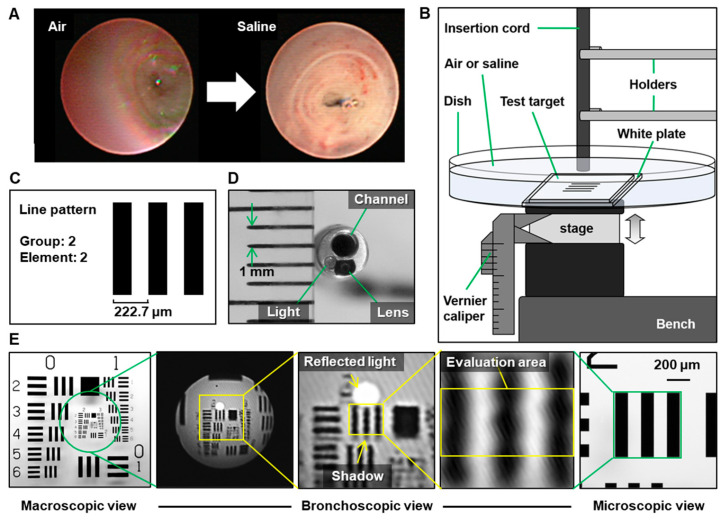
Equipment settings for evaluating endoscope resolution with or without saline. (**A**) Representative images of air vs. saline. (**B**) Schematic diagram of the equipment setting. (**C**) Test target and the line pattern for evaluating 111-μm resolution. (**D**) Image of the ultrathin bronchoscope. The distance between the green arrows on the ruler represents 1 mm. (**E**) Image of the line pattern on macroscopic (left), ultrathin bronchoscopic (center), and microscopic views (right, 4× lens). The yellow arrows indicate the reflected light and the shadow.

**Figure 2 diagnostics-15-03029-f002:**
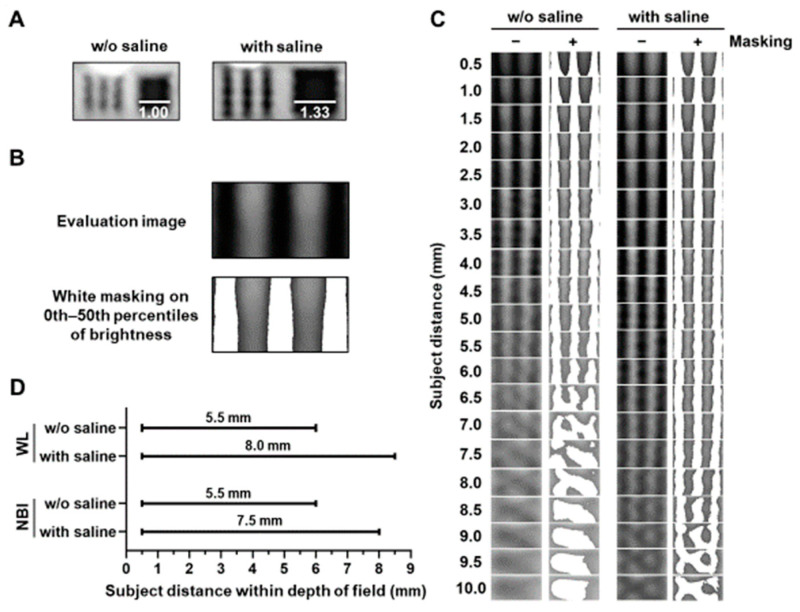
Evaluation of subject distance within the depth of field with or without saline. (**A**) Representative images obtained with or without saline at a subject distance (5.0 mm). Bars indicate the relative length. (**B**) Representative masking images of dark or bright dots in the evaluation image. (**C**) Saline increased the range of subject distance with separated white mask areas in evaluation images obtained using white-light (WL) imaging. (**D**) Saline increased the range of the subject distance within the depth of field using WL or narrow-band imaging (NBI).

**Figure 3 diagnostics-15-03029-f003:**
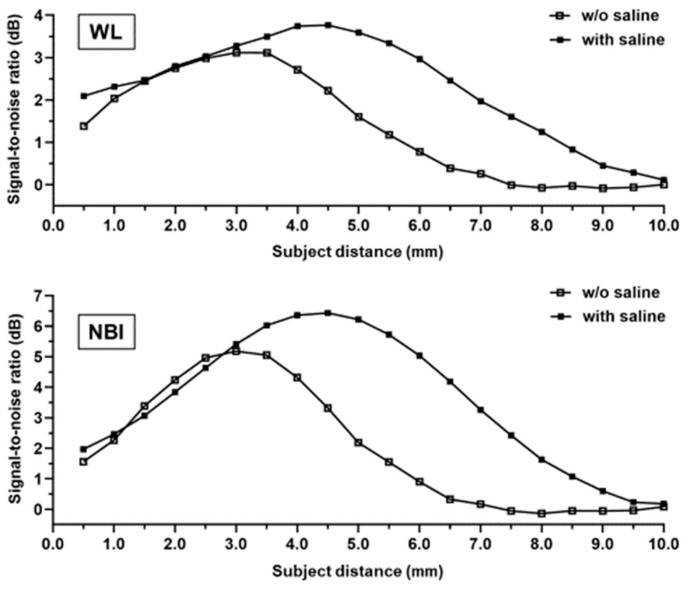
Evaluation of the signal-to-noise ratio. The distance between the endoscope and test target varies in terms of the signal-to-noise ratio in white-light (**top**) and narrow-band imaging (**bottom**).

**Figure 4 diagnostics-15-03029-f004:**
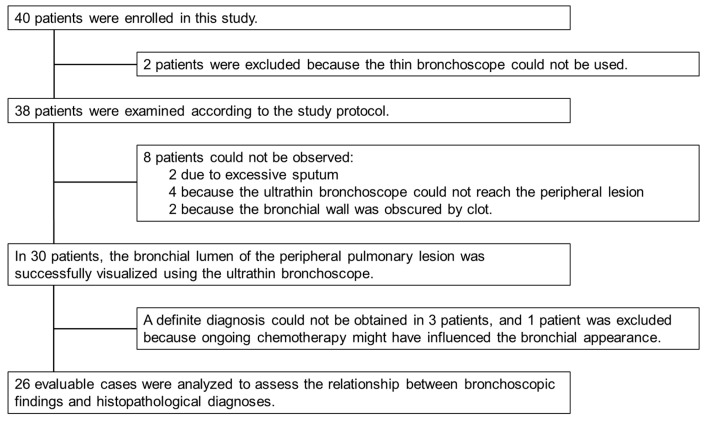
Study enrollment summary. Forty patients were enrolled in this study. Bronchoscopic findings of the peripheral lesions were observed in 30 patients using an ultrathin bronchoscope. One patient was excluded from the analysis because bronchoscopy was performed while the patient was receiving chemotherapy, which could have altered the intraluminal appearance of the lesion. The final diagnosis was confirmed, and we compared the bronchoscopic findings with the histopathological diagnoses in 26 patients.

**Figure 5 diagnostics-15-03029-f005:**
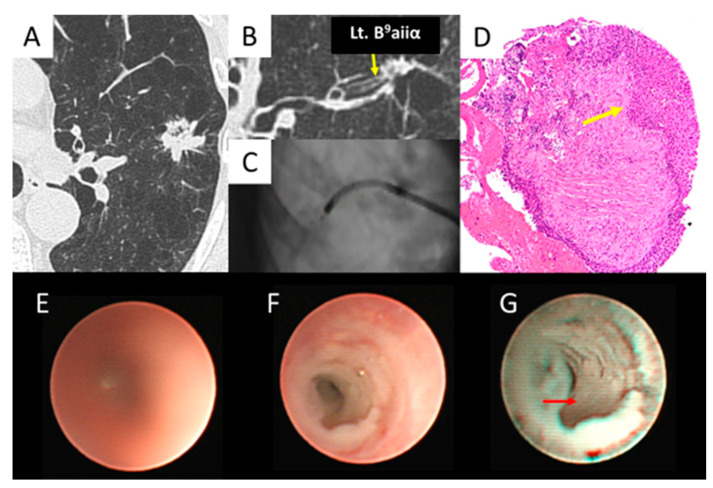
Epithelial type (squamous cell carcinoma). (**A**,**B**) Left B^9^aiiα (*yellow arrow*) entered the lesion. (**C**) The ultrathin bronchoscope reached the lesion. (**D**) The biopsy specimen obtained by thin bronchoscopy revealed squamous cell carcinoma (*yellow arrow*). (**E**) The ultrathin bronchoscope with a white light in air showed an unclear image of B^9^aiiα. (**F**) The ultrathin bronchoscope with white light in saline showed a whitish, irregular surface of B^9^aiiα. (**G**) The ultrathin bronchoscope with NBI in saline showed a whitish, irregular epithelium (*red arrow*) of B^9^aiiα.

**Figure 6 diagnostics-15-03029-f006:**
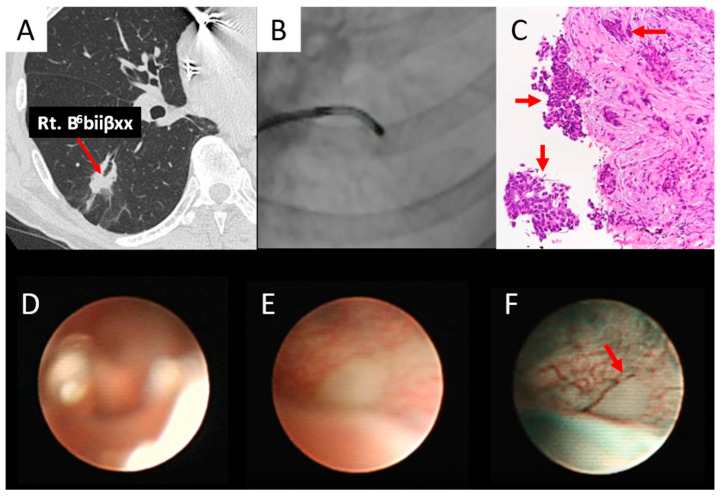
Subepithelial type (adenocarcinoma). (**A**) Right B^6^biiβxx (*red arrow*) entered the lesion. (**B**) The ultrathin bronchoscope reached the lesion. (**C**) The biopsy specimen obtained by thin bronchoscopy revealed adenocarcinoma (*red arrows*). (**D**) An ultrathin bronchoscope with white light in air showed an unclear image of B^6^biiβxx. (**E**) An ultrathin bronchoscope with white light in saline showed a whitish subepithelial lesion of B^6^biiβxx. (**F**) The ultrathin bronchoscope with NBI in saline showed a whitish subepithelial lesion and abrupt caliber change (red arrow) of the vessel of B^6^biiβxx.

**Table 1 diagnostics-15-03029-t001:** Characteristics of the patients and lesions.

Characteristics		*n* = 26
Age, median (range), years		74 (22–91)
Sex, *n* (%)	Male	20 (76.9)
	Female	6 (23.1)
Lesion located, *n*	Right upper lobe	8
	Right middle lobe	1
	Right lower lobe	2
	Left upper lobe	6
	Left lower lobe	9
Size of mass lesion, median (range), mm	Maximum diameter	23.3 (10.0–59.5)
	Minimum diameter	17.7 (7.0–40.0)
Bronchial inner diameter to enter the lesion, median (range), mm		1.3 (0.6–2.1)
Bronchus at the lesion *, median (range), bronchial generations		6 (3–8)
To reach the bronchus, median (range), bronchial generations	Ultrathin bronchoscope	5.5 (3–9)
	Thin bronchoscope	4 (2–7)
Examination time, median (range), min	By the ultrathin bronchoscope	15 (6–31)
	By the thin bronchoscope	20 (12–31)
	Total	38.5 (24–56)
Amount of saline injected into the bronchus, median (range), mL		13 (3–32) ^†^

* Bronchi at the target lesion on high-resolution computed tomography. ^†^ In one case, the injection quantity data for physiological saline were deficient.

**Table 2 diagnostics-15-03029-t002:** Bronchoscopic findings and histopathological diagnosis.

	Border	Form	Histopathological Diagnosis	*n*
Epithelial type (*n* = 4)	Clear	Flat	SCC	1
		Raised	Ad	1
	Unclear	Raised	SCC	1
	Invisible	Raised	SCC	1
Subepithelial type (*n* = 22)	Unclear	Flat	SCC	1
			Ad	8
			Adeno-squamous cell carcinoma	1
			Granulomatous inflammation	3
			Sarcoidosis	1
		Raised	Ad	3
			Small cell carcinoma	1
		Obstructive	SCC	1
			Ad	1
			Non-small cell carcinoma, not otherwise specified	2

SCC: squamous cell carcinoma. Ad: adenocarcinoma.

**Table 3 diagnostics-15-03029-t003:** Ultrathin bronchoscopic findings of raters 1 and 2 following air or saline injection in white light.

		% Distinguishable Cases		*p*-Value (Exact McNemar Test)
		Air	Saline	
Rater 1	The unevenness of the bronchial luminal surface	38.5	100.0	<0.0001
	Circular and/or longitudinal folds	38.5	92.3	0.00012
	Blood vessels of the subepithelium	53.8	96.2	0.00098
	Perspective	42.3	100.0	<0.0001
Rater 2	The unevenness of the bronchial luminal surface	7.7	42.3	0.0039
	Circular and/or longitudinal folds	42.3	92.3	0.00024
	Blood vessels of the subepithelium	0.0	42.3	0.00098
	Perspective	65.4	96.2	0.0078

Both raters were able to significantly evaluate the findings of the bronchial lumen following saline injection into the bronchus.

## Data Availability

The data that support the findings of this study are available on request from the corresponding author, [TO]. The data are not publicly available because the participants in this study did not provide written consent for their data to be shared publicly.

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
