# Peer review of "Bronchial Physiological Saline Injection to Visualize Peripheral Pulmonary Lesions in Ultrathin Bronchoscopy"

_diagnostics, 2025, doi:10.3390/diagnostics15233029_

Round 1

Reviewer 1 Report

Comments and Suggestions for Authors

This is a well-written manuscript. 

1. Have the authors considered the impact of patients with chronic bronchitis and structural lung diseases, as their mucus hypersecretion may serve as a confounding factor? How have these patients been accounted for in the study?

2. The authors should provide a clear discussion regarding the sample size calculation. 

    3.  The references primarily consist of older articles; new articles on the issue need to be cited.

Author Response

We sincerely appreciate the reviewers for their valuable comments and constructive suggestions, which have helped us to improve the quality of our manuscript. We have carefully revised the text according to their feedback, as detailed below.

Reviewer 1

Comment 1:
Have the authors considered the impact of patients with chronic bronchitis and structural lung diseases, as their mucus hypersecretion may serve as a confounding factor? How have these patients been accounted for in the study?

Response:
14 patients out of 40 in our cohort had symptoms of chronic bronchitis, and four had interstitial pneumonia. Because one of our inclusion criteria was that the bronchial lumen leading to the lesion must have an inner diameter of less than 4 mm, no patients with bronchiectasis were included. Given the limited sample size, we could not perform statistical analyses comparing the effect of saline injection between patients with and without chronic bronchitis. However, in our clinical experience, mucus often obscures peripheral bronchial visualization. Injecting saline helps to wash out the mucus and enables clearer observation of the distal bronchial lumen.
Corresponding changes: We added this information to the Discussion section.

Furthermore, in addition to its optical effects, saline also acts physically by displacing mucus, thereby enabling direct visualization of the bronchial lumen. This effect may be particularly beneficial when performing bronchoscopy in patients with chronic bronchitis.

Comment 2:
The authors should provide a clear discussion regarding the sample size calculation.

Response:
As there were no previous reports describing the relationship between ultrathin bronchoscopic intraluminal findings and histopathological types, a formal sample size calculation was not feasible. Therefore, this study was designed as an exploratory investigation. Based on the number of cases of solitary peripheral lung nodules diagnosed at Shimane University Hospital in 2017, we estimated that approximately 40 cases could be prospectively accumulated in one year and set this number as the target sample size.
Corresponding changes: We added this explanation to the Materials and Methods section.

Because no previous studies have reported the relationship between ultrathin bronchoscopic intraluminal findings and histopathological types, a formal sample size calculation was not feasible. Therefore, this study was designed as an exploratory investigation. Based on the number of cases of solitary peripheral lung nodules diagnosed at Shimane University Hospital in 2017, we estimated that approximately 40 cases could be prospectively enrolled within one year and set this number as the target sample size.

Comment 3:
The references primarily consist of older articles; new articles on the issue need to be cited.

Response:
We appreciate this comment and have added several recent references related to ultrathin bronchoscopy and newly developed devices (e.g., SUKEDACHI™ Broncho Dilation Balloon Catheter and Iriscope) to provide updated context.
Corresponding changes: The Discussion section was updated to include recent studies.

Reviewer 2 Report

Comments and Suggestions for Authors

1. Not enough relevant introduction given

2. How far can the scope advance with saline injection?

3. chemotherapy changed the bronchoscopic findings in one patient - What does it mean

4. Should compare the time needed than without saline injection

5. Limitation is no comparator group as all so we cannot tell if this technique really can increae the yield

6. May be more suitable to present as a case series

Author Response

We sincerely appreciate the reviewers for their valuable comments and constructive suggestions, which have helped us to improve the quality of our manuscript. We have carefully revised the text according to their feedback, as detailed below.

Reviewer 2

Comment 1:
Not enough relevant introduction given.

Response:
We agree with the reviewer’s insight. We have expanded the Introduction to clarify the background of the study and our aims.

Corresponding changes: The following sentences were added to the introduction section.

A future diagnostic pathway is expected to be established in which the intraluminal findings of bronchi containing peripheral pulmonary lesions are directly observed using an ultrathin bronchoscope, enabling real-time differentiation between benign and malignant lesions and facilitating targeted biopsies under direct vision.

Comment 2:
How far can the scope advance with saline injection?

Response:
The ultrathin bronchoscope was advanced as far as possible until resistance was encountered, and saline was injected at that position. The bronchoscope was not advanced further after the saline injection.
Corresponding changes: Clarified in the Materials and Methods section.

The ultrathin bronchoscope was advanced as far as possible until resistance was encountered, and saline was injected at that position. The bronchoscope was not advanced further after the saline injection.

Comment 3:
“Chemotherapy changed the bronchoscopic findings in one patient”—What does it mean?

Response:
One patient underwent bronchoscopy for re-biopsy while receiving chemotherapy for lung cancer. Because chemotherapy may have altered the bronchial intraluminal appearance, this case was excluded from the analysis.
Corresponding changes: We Clarified in the Results section.

We excluded four patients. In three patients, a definite diagnosis could not be obtained. One patient was excluded because the bronchoscopy was performed during on-going chemotherapy, which might have affected the bronchoscopic findings.

We also clarified in the Figure 4. 

, and 1 patient was excluded because ongoing chemotherapy might have influenced the bronchial appearance.

We clarified in the legend of the Figure 4.

One patient was excluded from the analysis because bronchoscopy was performed while the patient was receiving chemotherapy, which could have altered the intraluminal appearance of the lesion.

Comment 4:
Should compare the time needed than without saline injection.

Response:
In this study, all cases involved saline injection; therefore, we could not directly compare the procedure time with cases without saline injection. However, saline injection takes only a few seconds, and we believe it does not prolong the total examination time.
Corresponding changes: Noted in the Discussion section.

Moreover, saline injection takes only a few seconds, and we believe it does not prolong the total examination time.

Comment 5:
Limitation is no comparator group as all so we cannot tell if this technique really can increase the yield.

Response:
We agree with the reviewer’s observation. It was not possible to assess whether saline injection improved the cytological or histological diagnostic yield in this study. We have added this point to the Limitations section, emphasizing the need for further comparative studies to evaluate its diagnostic utility. We also mentioned the emergence of new devices such as Iriscope and SUKEDACHI™, which warrant future comparative evaluation.
Corresponding changes: Added to the Discussion section.

This study could not evaluate whether saline injection improved the cytological or histological diagnostic yield. Further comparative studies are therefore required to clarify its diagnostic utility. In addition, newly developed devices such as the Iriscope and SUKEDACHI™ warrant future comparative evaluation to determine their relative advantages.

Comment 6:
May be more suitable to present as a case series.

Response:
We appreciate this suggestion. However, in addition to basic experiments, our study included 26 clinical cases with detailed quantitative data such as the amount of saline injected, the number of bronchial generations near the peripheral lesion, and the bronchial generations reached by the ultrathin and thin bronchoscopes. We believe these data provide sufficient analytical depth for an original article rather than a case series.

Round 2

Reviewer 2 Report

Comments and Suggestions for Authors
  1. Why authors think adding 1 short parargraph is considered enough when I said it is too brief?
  2. The ultrathin bronchoscope was advanced as far as possible - How many cm?
  3. Really have luminal changes with chemo? Any evidence to support?

Round 3

Reviewer 2 Report

Comments and Suggestions for Authors

Thanks for the revision